# COLLAPSE OF DEEP AND NARROW NEURAL NETS

## ABSTRACT

Recent theoretical work has demonstrated that deep neural networks have superior performance over shallow networks, but their training is more difficult, e.g., they suffer from the vanishing gradient problem. This problem can be typically resolved by the rectified linear unit (ReLU) activation. However, here we show that even for such activation, deep and narrow neural networks (NNs) will converge to erroneous mean or median states of the target function depending on the loss with high probability. Deep and narrow NNs are encountered in solving partial differential equations with high-order derivatives. We demonstrate this collapse of such NNs both numerically and theoretically, and provide estimates of the probability of collapse. We also construct a diagram of a safe region for designing NNs that avoid the collapse to erroneous states. Finally, we examine different ways of initialization and normalization that may avoid the collapse problem. Asymmetric initializations may reduce the probability of collapse but do not totally eliminate it.

## 1 INTRODUCTION

The best-known universal approximation theorems of neural networks (NNs) were obtained almost three decades ago by Cybenko (1989) and Hornik et al. (1989), stating that every measurable function can be approximated accurately by a single-hidden-layer neural network, i.e., a shallow neural network. Although powerful, these results do not provide any information on the required size of a neural network to achieve a pre-specified accuracy. In Barron (1993), the author analyzed the size of a neural network to approximate functions using Fourier transforms. Subsequently, in Mhaskar (1996), the authors considered optimal approximations of smooth and analytic functions in shallow networks, and demonstrated that $\epsilon^{-d/n}$ neurons can uniformly approximate any $C^n$-function on a compact set in $\mathbb{R}^d$ with error $\epsilon$. This is an interesting result and it shows that to approximate a three-dimensional function with accuracy $10^{-6}$ we need to design a NN with $10^{18}$ neurons for a $C^1$ function, but for a very smooth function, e.g., $C^6$, we only need 1000 neurons. In the last 15 years, deep neural networks (i.e., networks with a large number of layers) have been used very effectively in diverse applications.

After some initial debate, at the present time, it seems that deep NNs perform better than shallow NNs of comparable size, e.g., a 3-layer NN with 10 neurons per layer may be a better approximator than a 1-layer NN with 30 neurons. From the approximation point of view, there are several theoretical results to explain this superior performance. In Eldan & Shamir (2016), the authors showed that a simple approximately radial function can be approximated by a small 3-layer feed-forward NN, but it cannot be approximated by any 2-layer network with the same accuracy irrespective of the activation function, unless its width is exponential in the dimension (see Mhaskar et al. (2017); Mhaskar & Poggio (2016); Delalleau & Bengio (2011); Poggio et al. (2017) for further discussions). In Liang & Srikant (2017) (see also Yarotsky (2017)), the authors claimed that for $\epsilon$-approximation of a large class of piecewise smooth functions using the rectified linear unit (ReLU) $\max(x, 0)$ activation function, a multilayer NN using $\Theta(\log(1/\epsilon))$ layers only needs $\mathcal{O}(\text{poly} \log(1/\epsilon))$ neurons, while $\Omega(\text{poly}(1/\epsilon))$ neurons are required by NNs with $o(\log(1/\epsilon))$ layers. That is, the number of neurons required by a shallow network to approximate a function is exponentially larger than the corresponding number of neurons needed by a deep network for a given accuracy level of function approximation. In Petersen & Voigtlaender (2018), the authors studied approximation theory of a class of (possibly discontinuous) piecewise $C^\beta$ functions for ReLU NN, and they found that no more than $\mathcal{O}(\epsilon^{-2(d-1)/\beta})$ nonzero weights are required to approximate the function in the $L^2$ sense,

which proves to be optimal. Under this optimality condition, they also show that a minimum depth (up to a multiplicative constant) is given by $\beta/d$ to achieve optimal approximation rates. As for the expressive power of NNs in terms of the width, Lu et al. (2017) showed that any Lebesgue integrable function from $\mathbb{R}^d$ to $\mathbb{R}$ can be approximated by a ReLU forward NN of width $d + 4$ with respect to $L^1$ distance, and cannot be approximated by any ReLU NN whose width is no more than $d$. Hanin & Sellke (2017) showed that any continuous function can be approximated by a ReLU forward NN of width $d_{in} + d_{out}$, and they also give a quantitative estimate of the depth of the NN; here $d_{in}$ and $d_{out}$ are the dimensions of the input and output, respectively. For classification problems, networks with a pyramidal structure and a certain class of activation functions need to have width larger than the input dimension in order to produce disconnected decision regions (Nguyen et al., 2018).

With regards to optimum activation function employed in the NN approximation, before 2010 the two commonly used non-linear activation functions were the logistic sigmoid $1/(1 + e^{-x})$ and the hyperbolic tangent ($\texttt{tanh}$); they are essentially the same function by simple re-scaling, i.e., $\tanh(x) = 2\,\text{sigmoid}(2x) - 1$. The deep neural networks with these two activations are difficult to train (Glorot & Bengio, 2010). The non-zero mean of the sigmoid induces important singular values in the Hessian (LeCun et al., 1998), and they both suffer from the vanishing gradient problem, especially through neurons near saturation (Glorot & Bengio, 2010). In 2011, ReLU was proposed, which avoids the vanishing gradient problem because of its linearity, and also results in highly sparse NNs (Glorot et al., 2011). Since then, ReLU and its variants including leaky ReLU (LReLU) (Maas et al., 2013), parametric ReLU (PReLU) (He et al., 2015) and ELU (Clevert et al., 2015) are favored in almost all deep learning models. Thus, in this study, we focus on the ReLU activation.

While the aforementioned theoretical results are very powerful, they do not necessarily coincide with the results of training of NNs in practice which is NP-hard (Šíma, 2002). For example, while the theory may suggest that the approximation of a multi-dimensional smooth function is accurate for NN with 10 layers and 5 neurons per layer, it may not be possible to realize this NN approximation in practice. Fukumizu & Amari (2000) first proved that existence of local minima poses a serious problem in learning of NNs. After that, more work has been done to understand bad local minima under different assumptions (Zhou & Liang, 2017; Du et al., 2017; Safran & Shamir, 2017; Wu et al., 2018; Yun et al., 2018). Besides local minima, singularity (Amari et al., 2006) and bad saddle points (Kawaguchi, 2016) also affect training of NNs. Our paper focuses on a particular kind of bad local minima, i.e., those encountered in deep and narrow neural networks collapse with high probability. This is the topic of our work presented in this paper. Our results are summarized in Fig. 6, which shows a diagram of the safe region of training to achieve the theoretically expected accuracy. As we show in the next section through numerical simulations as well as in subsequent sections through theoretical results, there is very high probability that for deep and narrow ReLU NNs will converge to an erroneous state, which may be the mean value of the function or its partial mean value. However, if the NN is trained with proper normalization techniques, such as batch normalization (Ioffe & Szegedy, 2015), the collapse can be avoided. Not every normalization technique is effective, for example, weight normalization (Salimans & Kingma, 2016) leads to the collapse of the NN.

## 2 COLLAPSE OF DEEP AND NARROW NEURAL NETWORKS

In this section, we will present several numerical tests for one- and two-dimensional functions of different regularity to demonstrate that deep and narrow NNs collapse to the mean value or partial mean value of the function.

It is well known that it is hard to train deep neural networks. Here we show through numerical simulations that the situation gets even worse if the neural networks is narrow. First, we use a 10-layer ReLU network with width 2 to approximate $y(x) = |x|$, and choose the mean squared error (MSE) as the loss. In fact, $y(x)$ can be represented exactly by a 2-layer ReLU NN with width 2, $|x| = \text{ReLU}(x) + \text{ReLU}(-x) = \begin{bmatrix} 1 & 1 \end{bmatrix} \text{ReLU}(\begin{bmatrix} 1 \\ -1 \end{bmatrix} x)$. However, our numerical tests show that there is a high probability ($\sim 90\%$) for the NN to collapse to the mean value of $y(x)$ (Fig. 1), no matter what kernel initializers (He normal (He et al., 2015), LeCun normal (LeCun et al., 1998; Klambauer et al., 2017), Glorot uniform (Glorot & Bengio, 2010)) or optimizers (first order or second order including SGD, SGDNesterov (Sutskever et al., 2013), AdaGrad (Duchi et al., 2011),

AdaDelta (Zeiler, 2012), RMSProp (Hinton, 2014), Adam (Kingma & Ba, 2015), BFGS (Nocedal & Wright, 2006), L-BFGS (Byrd et al., 1995)) are employed. The training data were sampled from a uniform distribution on $[-\sqrt{3}, \sqrt{3}]$, and the minibatch size was chosen as 128 during training. We find that when this happens, in most cases the bias in the last layer is the mean value of the function $y(x)$, and the composition of all the previous layers is equivalent to a zero function. It can be proved that under these conditions, the gradient vanishes, i.e., the optimization stops (Corollary 5). For functions of different regularity, we observed the same collapse problem, see Fig. 2 for the $C^\infty$ function $y(x) = x\sin(5x)$ and Fig. 3 for the $L^2$ function $y(x) = 1_{\{x>0\}} + 0.2\sin(5x)$.

For multi-dimensional inputs and outputs, this collapse phenomenon is also observed in our simulations. Here, we test the target function $\mathbf{y}(\mathbf{x})$ with $d_{in} = 2$ and $d_{out} = 2$, which can be represented by a 2-layer neural network with width 4, $\mathbf{y}(\mathbf{x}) = \begin{bmatrix} |\mathbf{x}_1 + \mathbf{x}_2| \\ |\mathbf{x}_1 - \mathbf{x}_2| \end{bmatrix} = \begin{bmatrix} 1 & 1 & & \\ & & 1 & 1 \end{bmatrix} \text{ReLU}(\begin{bmatrix} 1 & 1 \\ -1 & -1 \\ 1 & -1 \\ -1 & 1 \end{bmatrix} \mathbf{x})$. When training a 10-layer ReLU network with width 4, there is a very high probability for the NN to collapse to the mean value or with low probability to the partial mean value of $\mathbf{y}(\mathbf{x})$ (Fig. 4).

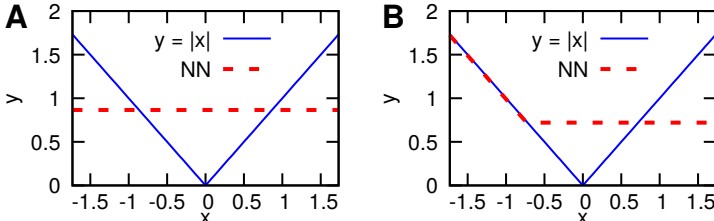

Figure 1: Demonstration of the neural network collapse to the mean value (**A**, with very high probability) or the partial mean value (**B**, with low probability) for the $C^0$ target function $y(x) = |x|$. The gradient vanishes in both cases (see Corollaries 5 and 6). A 10-layer ReLU neural network with width 2 is employed in both (**A**) and (**B**). The biases are initialized to 0, and the weights are randomly initialized from a symmetric distribution. The loss function is MSE.

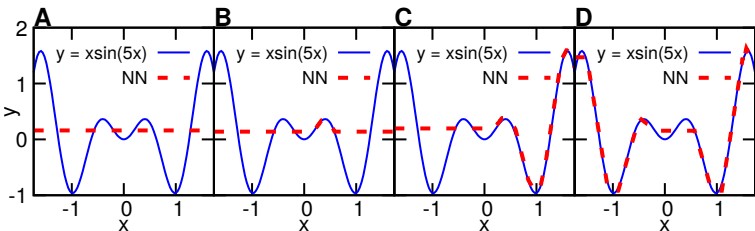

Figure 2: Similar behavior for the $C^\infty$ target function $y(x) = x\sin(5x)$. The network parameters, loss function, and initializations are the same as in Fig. 1. (**A**) corresponds to the mean value of the target function with high probability. (**B**, **C**, **D**) correspond to partial mean values with low probability and are induced by different random initializations.

We also observed the same collapse problem for other losses, such as the mean absolute error (MAE); the results are summarized in Fig. 5 for three different functions with varying regularity. Furthermore, we find that for MSE loss, the constant is the mean value of the target function, while for MAE it is the median value.

## 3 INITIALIZATION OF RELU NETS

As we demonstrated above, when the weights of the ReLU NN are randomly initialized from a symmetric distribution, the deep and narrow NN will collapse with high probability. This type of

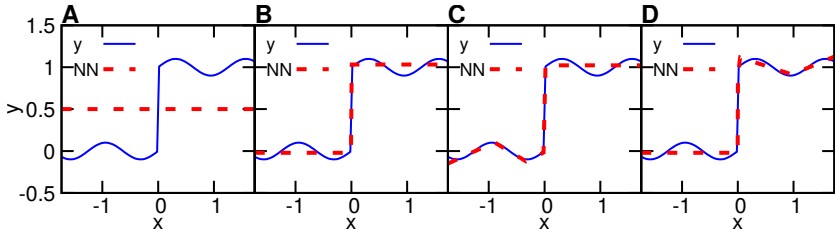

Figure 3: Similar behavior for the $L^2$ target function $y(x) = 1_{\{x>0\}} + 0.2\sin(5x)$. The network parameters, loss function, and initializations are the same as in Fig. 1. (**A**) corresponds to the mean value of the target function with high probability. (**B**, **C**, **D**) correspond to partial mean values with low probability and are induced by different random initializations.

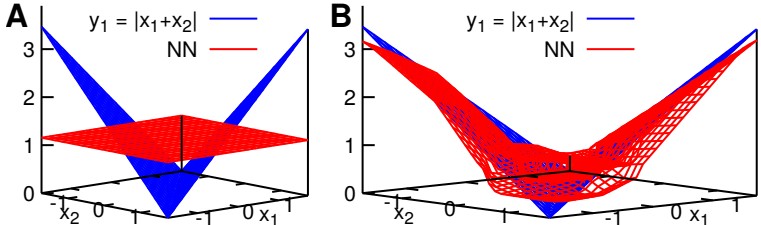

Figure 4: Demonstration of the neural network collapse to the mean value (**A**, with very high probability) or the partial mean value (**B**, with low probability) for the $C^0$ 2-dimensional (vector) target function $\mathbf{y}(\mathbf{x}) = [|\mathbf{x}_1 + \mathbf{x}_2|, |\mathbf{x}_1 - \mathbf{x}_2|]$. The gradient vanishes in both cases (see Corollaries 5 and 6). A 10-layer ReLU neural network with width 4 is employed in both (**A**) and (**B**). The biases are initialized to 0, and the weights are initialized from a symmetric distribution. The loss function is MSE.

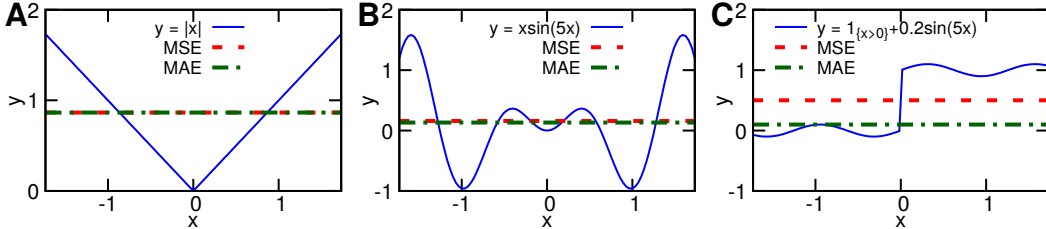

Figure 5: Effect of the loss function on the behavior of the collapse of the neural network. MSE (used in Figs 1, 2, 3) is compared against the MAE. The collapse of the NN is independent of the loss function (see Theorem 4).

initialization is widely used in real applications. Here, we demonstrate that this initialization avoids the problem of exploding/vanishing mean activation length, therefore this is beneficial for training neural networks.

We study a feed-forward neural network $\mathcal{N} : \mathbb{R}^{d_{in}} \to \mathbb{R}^{d_{out}}$ with $L$ layers and $N^l$ neurons in the layer $l$ ($N^0 = d_{in}$, $N^L = d_{out}$). The weights and biases in the layer $l$ are an $N^l \times N^{l-1}$ weight matrix $\mathbf{W}^l$ and $\mathbf{b}^l \in \mathbb{R}^{N^l}$, respectively. The input is $\mathbf{x}^0 \in \mathbb{R}^{d_{in}}$, and the neural activity in the layer $l$ is $\mathbf{x}^l \in \mathbb{R}^{N^l}$. The feed-forward dynamics is given by

$$\mathbf{x}^l = \phi(\mathbf{h}^l) \qquad \mathbf{h}^l = \mathbf{W}^l \mathbf{x}^{l-1} + \mathbf{b}^l \quad \text{for } l = 1, \dots, L-1,$$

$$\mathcal{N}(\mathbf{x}^0) \equiv \mathbf{x}^L = \mathbf{h}^L = \mathbf{W}^L \mathbf{x}^{L-1} + \mathbf{b}^L,$$

where $\phi$ is a component-wise activation function.

Following the work in Poole et al. (2016), we investigate how the length of the input propagates through neural networks. The normalized squared length of the vector before activation at each layer is defined as

$$q^l = \frac{1}{N_l} \sum_{i=1}^{N_l} (\mathbf{h}_i^l)^2, \tag{1}$$

where $\mathbf{h}_i^l$ denotes the entry $i$ of the vector $\mathbf{h}^l$. If the weights and biases are drawn i.i.d. from a zero mean Gaussian with variance $\sigma_w^2/N^{l-1}$ and $\sigma_b^2$ respectively, then the length at layer $l$ can be obtained from its previous layer (see the proof in the appendix of (Poole et al., 2016), which we include here in Appendix A)

$$\mathbb{E}[q^l] = \sigma_w^2 \int \mathcal{D}z \phi(\sqrt{\mathbb{E}[q^{l-1}]}z)^2 + \sigma_b^2, \quad \text{for } l \geq 2, \tag{2}$$

where $\mathcal{D}z = \frac{dz}{\sqrt{2\pi}} e^{-\frac{z^2}{2}}$ is the standard Gaussian measure, and the initial condition is $\mathbb{E}[q^1] = \sigma_w^2 q^0 + \sigma_b^2$, $q^0 = \frac{1}{N_0} \mathbf{x}^0 \cdot \mathbf{x}^0$. When $\phi$ is ReLU, the recursion is simplified to

$$\mathbb{E}[q^l] = \sigma_w^2 \int \mathcal{D}z \text{ReLU}(\sqrt{\mathbb{E}[q^{l-1}]}z)^2 + \sigma_b^2 = \sigma_w^2 \int_0^\infty \mathcal{D}z(\sqrt{\mathbb{E}[q^{l-1}]}z)^2 + \sigma_b^2$$

$$= \sigma_w^2 \mathbb{E}[q^{l-1}] \int_0^\infty z^2 \mathcal{D}z + \sigma_b^2 = \frac{\sigma_w^2}{2} \mathbb{E}[q^{l-1}] \int_{-\infty}^\infty z^2 \mathcal{D}z + \sigma_b^2 = \frac{\sigma_w^2}{2} \mathbb{E}[q^{l-1}] + \sigma_b^2. \tag{3}$$

For ReLU, He normal (He et al., 2015), i.e., $\sigma_w^2 = 2$ and $\sigma_b = 0$, is widely used. This choice guarantees that $\mathbb{E}[q^l] = \mathbb{E}[q^{l-1}]$, which neither shrinks nor expands the inputs. In fact, this result explains the success of He normal in applications. A parallel work by Hanin & Rolnick (2018) shows that initializing weights from a symmetric distribution with variance 2/fan-in (fan-in is the dimension of the input of each layer) avoids the problem of exploding/vanishing mean activation length. Here we arrived at the same conclusion but with much less work.

## 4 THEORETICAL ANALYSIS OF THE COLLAPSE PROBLEM

In this section, we present the theoretical analysis of the collapse behavior observed in Section 2, and we also derive an estimate of the probability of this collapse. We start by stating the following assumptions for a ReLU feed-forward neural network $\mathcal{N}(\mathbf{x}^0) : \mathbb{R}^{d_{in}} \to \mathbb{R}^{d_{out}}$ with $L$ layers and $N^l$ neurons in the layer $l$ ($N^0 = d_{in}$, $N^L = d_{out}$):

A1 The domain $\Omega \subset \mathbb{R}^{d_{in}}$ for $\mathcal{N}$ is a connected space with at least two points;

A2 The weight matrix $\mathbf{W}^l \in \mathbb{R}^{N^l \times N^{l-1}}$ of any layer $l \in \{1, 2, \dots, L\}$ is a random matrix, where the joint distribution of $(\mathbf{W}_{i1}^l, \mathbf{W}_{i2}^l, \dots, \mathbf{W}_{iN^{l-1}}^l)$ is absolutely continuous with respect to Lebesgue measure for $i = 1, 2, \dots, N^l$.

**Remark**: We point out here that the connectedness in assumption A1 is a very weak requirement for the input space. The weights in a neural network are usually sampled independently from continuous distributions in real applications, and thus the assumption A2 is satisfied at the NN initialization

stage; during training, the assumption A2 is usually maintained due to stochastic gradients of mini-batch.

**Lemma 1.** *With assumptions A1 and A2, if $\mathcal{N}(\mathbf{x}^0)$ is a constant function, then there exists a layer $l \in \{1, \ldots, L-1\}$ such that $\mathbf{h}^l \leq \mathbf{0}^1$ and $\mathbf{x}^l = \mathbf{0}$ $\forall \mathbf{x}^0 \in \Omega$, with probability 1 (wp1).*

**Corollary 2.** *With assumptions A1 and A2, if $\mathcal{N}(\mathbf{x}^0)$ is bias-free and a constant function, then there exists a layer $l \in \{1, \ldots, L-1\}$ such that for any $n \geq l$, it holds $\mathbf{h}^n \leq \mathbf{0}$ and $\mathbf{x}^n = \mathbf{0}$ wp1.*

**Lemma 3.** *With assumptions A1 and A2, if $\mathcal{N}(\mathbf{x}^0)$ is a constant function, then any order gradients of the loss function with respect to the weights and biases in layers $1, \ldots, l$ vanish, where $l$ is the layer obtained in Lemma 1.*

**Theorem 4.** *For a ReLU feed-forward neural network $\mathcal{N}(\mathbf{x}^0)$ with assumption A1, if the assumption A2 is satisfied during the initialization, and there exists a layer $l$ such that $\mathbf{x}^l(\mathbf{x}^0) \equiv \mathbf{0}$ for any input $\mathbf{x}^0$, then for any function $\mathbf{y}(\mathbf{x}^0)$ and $\mathbf{x}^0 \in \Omega$, $\mathcal{N}$ is eventually optimized to a constant function when training by a gradient based optimizer. If using $L^2$ loss and $\mathbb{E}_{\mathbf{x}^0}[\mathbf{y}(\mathbf{x}^0)]$ exists, then the resulted constant is $\mathbb{E}_{\mathbf{x}^0}[\mathbf{y}(\mathbf{x}^0)]$, which we write as $\mathbb{E}[\mathbf{y}]$ if no confusion arises; if using $L^1$ loss and the median of the distribution of $\mathbf{y}$ exists, then the resulted constant is the median.*

**Remark**: See Appendices B, C, D and E for the proofs of Lemma 1, Corollary 2, Lemma 3 and Theorem 4, respectively. MAE and MSE loss used in practice are discrete versions of $L^1$ and $L^2$ loss, respectively, if the size of minibatch is large.

**Corollary 5.** *With assumptions A1 and A2, for a ReLU feed-forward neural network $\mathcal{N}(\mathbf{x}^0)$ and any bounded function $\mathbf{y}(\mathbf{x}^0)$, $\mathbf{x}^0 \in \Omega$, if $\mathcal{N}(\mathbf{x}^0)$ is a constant function with the value $\mathbb{E}[\mathbf{y}]$, then the gradients of the loss function with respect to any weight or bias vanish when using the $L^2$ loss.*

Corollary 5 can be generalized to the following corollary including more general converged mean states.

**Corollary 6.** *With assumptions A1 and A2, for a ReLU feed-forward neural network $\mathcal{N}(\mathbf{x}^0)$ and any bounded function $\mathbf{y}(\mathbf{x}^0)$, $\mathbf{x}^0 \in \Omega$, if $\exists K_1, \ldots, K_n \subset \Omega$ and each $K_i$ is a connected domain with at least two points, such that*

$$\mathcal{N}(\mathbf{x}^0) = \begin{cases} \mathbf{y}(\mathbf{x}^0) & \mathbf{x}^0 \in \Omega \setminus \cup_{i=1}^n K_i \\ \mathbb{E}_{\mathbf{x}_{K_i}^0}[\mathbf{y}(\mathbf{x}_{K_i}^0)] & \mathbf{x}^0 \in K_i \quad for \ i = 1, \ldots, n \end{cases},$$

*then the gradients of the loss function with respect to any weight or bias vanish when using the $L^2$ loss. Here $\mathbf{x}_{K_i}^0$ is the random variable of $\mathbf{x}^0$ restricted to $K_i$.*

See Appendices F and G for the proofs of Corollaries 5 and 6. We can see that Corollary 5 is a special case of Corollary 6 with $\cup_{i=1}^n K_i = \Omega$.

**Lemma 7.** *Let us assume that a one-layer ReLU feed-forward neural network $\mathcal{N}_1$ is initialized independently by symmetric nonzero distributions, i.e., any weight or bias of $\mathcal{N}_1$ is initialized by a symmetric nonzero distribution, which can be different for different parameters. Then, for any fixed input the corresponding output is zero with probability $(1/2)^{d_{out}}$, except the special case where all biases and the input are zero yielding that the output is always zero.*

**Theorem 8.** *If a ReLU feed-forward neural network $\mathcal{N}$ with $L$ layers assembled width $N^1, \ldots, N^L$ is initialized randomly by symmetric nonzero distributions for weights and zero biases, then for any fixed nonzero input, the corresponding output is zero with probability $1 - \Pi_{l=1}^L(1 - (1/2)^{N^l})$ if the last layer also employs ReLU activation, otherwise with the probability $1 - \Pi_{l=1}^{L-1}(1 - (1/2)^{N^l})$.*

See Appendices H and I for the proofs of Lemma 7 and Theorem 8. Although biases are initialized to 0 in most applications, for the sake of completeness, we also consider the case where biases are not initialized to 0.

**Proposition 9.** *If a ReLU feed-forward neural network $\mathcal{N}$ with $L$ layers assembled width $N^1, \ldots, N^L$ is initialized randomly by symmetric nonzero distributions (weights and biases), then for any fixed nonzero input, the corresponding output is zero with probability $(1/2)^{N^L}$ if the last layer also employs ReLU activation, otherwise the output is equal to the last bias $\mathbf{b}^L$ with probability $(1/2)^{N^{L-1}}$.*

---

[1]$\mathbf{a} \leq \mathbf{b}$ denotes $\mathbf{a}_i \leq \mathbf{b}_i$ for any index $i$, i.e., component-wise. Similarly for $<, >$ and $\geq$.

See Appendix J for the proof of Proposition 9. We note that Theorem 8 provides the probability for any given input, but in Theorem 4 it requires that the entire neural network is a zero function. Hence, the probability in Theorem 8 is an upper bound. In the following theorem, we give a theoretical formula of the probability for the NN with width 2.

**Proposition 10.** *Suppose the origin is an interior point of $\Omega$. Consider a bias-free ReLU neural network with $d_{in} = 1$, width 2 and $L$ layers, and weights are initialized randomly by symmetric nonzero distributions. Then for this neural network, the probability of being initialized to a constant function is the last component of $\pi^L$, where*

$$\pi^L = P^{L-1}\pi^1, \tag{4}$$

*with $\pi^1$ and $P$ being the probability distribution after the first layer and the probability transition matrix when one more layer is added, respectively. Here every layer employs the ReLU activation.*

See Appendix K for the derivation of $\pi^1$ and $P$. For general cases, we found that it is hard to obtain an explicit expression for the probability, so we used numerical simulations instead, where 1 million samples of random initialization are used to calculate each probability estimation. We show both theoretically (Theorem 8, Propositions 9 and 10) and numerically that NN has the same probability to collapse no matter what symmetric distributions are used, even if different distributions are used for different weights. On the other hand, to keep the collapse probability less than $p$, because the probability obtained in Theorem 8 is an upper bound, which corresponds to a safer maximum number of layers, we have that $1 - \Pi_{l=1}^{L}(1 - (1/2)^N) \leq p$, which implies the upper bound of the depth of NN

$$L \leq \frac{\ln(1-p)}{\ln(1-(1/2)^N)}. \tag{5}$$

Theorem 8 shows that when the NN gets deeper and narrower, the probability of the NN initialized to a zero function is higher (Fig. 6A). Hence, we have higher probability of vanishing gradient in almost all the layers, rather than just some neurons. In our experiments, we also found that there is very high probability that the gradient is 0 for all parameters except in the last layer, because ReLU is not used in the last layer. During the optimization, the neural network thus can only optimize the parameters in the last layer (Theorem 4). When we design a neural network, we should keep the probability less than 1% or 10%. As a practical guide, we constructed a diagram shown in Fig. 6B that includes both theoretical predictions and our numerical tests. We see that as the number of layers increases, the numerical tests match closer the theoretical results. It is clear from the diagram that a 10-layer NN of width 10 has a probability of only 1% to collapse whereas a 10-layer NN of width 5 has a probability greater than 10% to collapse; for width of three the probability is greater than 60%.

## 5 TRAINING DEEP AND NARROW NEURAL NETWORKS

In this section, we present some training techniques and examine which ones do not suffer from the collapse problem.

### 5.1 ASYMMETRIC WEIGHT INITIALIZATION

Our analysis applies for any symmetric initialization, so it is straightforward to consider asymmetric initializations. The asymmetric initializations proposed in the literature include orthogonal initialization (Saxe et al., 2014) and layer-sequential unit-variance (LSUV) initialization (Mishkin & Matas, 2016). LSUV is the orthogonal initialization combined with rescaling of weights such that the output of each layer has unit variance. Because weight rescaling cannot make the output escape from the negative part of ReLU, it is sufficient to consider the orthogonal initialization. The probability of collapse when using orthogonal initialization is very close to and a little lower than that when using symmetric distributions (Fig. 7). Therefore, orthogonal initialization cannot treat the collapse problem.

### 5.2 NORMALIZATION AND DROPOUT

As we have shown in the previous section, deep and narrow neural networks cannot be trained well directly with gradient-based optimizers. Here, we employ several widely used normalization

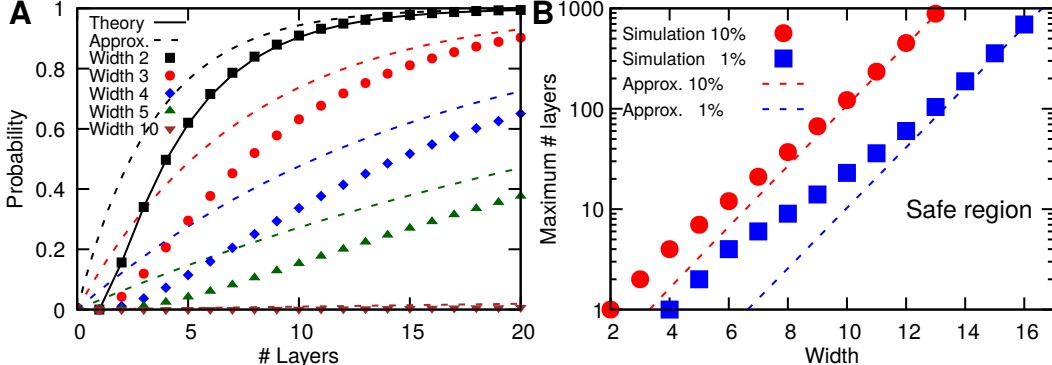

Figure 6: Probability of a ReLU NN to collapse and the safe operating region. (**A**) Probability of NN to collapse as a function of the number of layers for different widths. The solid black line represents the theoretical probability (Proposition 10). The dash lines represent the approximated probability (Theorem 8). The symbols represent our numerical tests. Similar colors correspond to the same width. A ReLU feed-forward NN is more likely to become a zero function when it is deeper and narrower. A bias-free ReLU feed-forward NN with $d_{in} = 1$ is employed with weights randomly initialized from symmetric distributions. (The last layer also applies activations.) (**B**) Diagram indicating safe operating regions for a ReLU NN. The dash lines represent Eq. 5 based on Theorem 8 while the symbols represent our numerical tests. The maximum number of layers of a neural network can be used at different width to keep the probability of collapse less than 1% or 10%. The region below the blue line is the safe region when we design a neural network. As the width increases the theoretical predictions match closer with our numerical simulations.

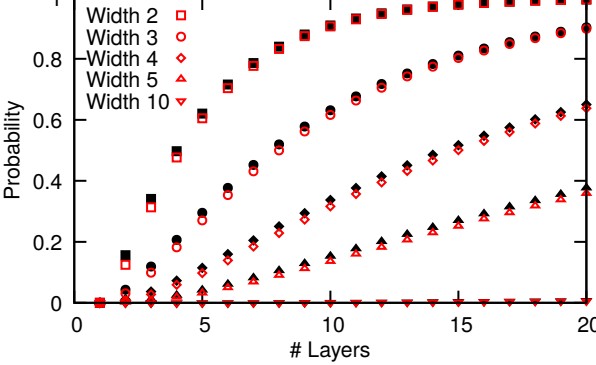

Figure 7: Effect of initialization on the collapse of NN. Plotted is the probability of collapse of a bias-free ReLU NN with $d_{in} = 1$ with different width and number of layers. The black filled symbols correspond to symmetric initialization while the red open symbols correspond to orthogonal initialization.

techniques to train this kind of networks. We do not consider some methods, such as Highway (Srivastava et al., 2015) and ResNet (He et al., 2016), because in these architectures the neural nets are no longer the standard feed-forward neural networks. Current normalization methods mainly include batch normalization (BN) (Ioffe & Szegedy, 2015), layer normalization (LN) (Ba et al., 2016), weight normalization (WN) (Salimans & Kingma, 2016), instance normalization (IN) (Ulyanov et al., 2016), group normalization (GN) (Wu & He, 2018), and scaled exponential linear units (SELU) (Klambauer et al., 2017). BN, LN, IN and GN are similar techniques and follow the same formulation, see Wu & He (2018) for the comparison.

Because we focus on the performance of these normalization methods on narrow nets and the width of the neural network must be larger than the dimension of the input to achieve a good approximation, we only test the normalization methods on low dimensional inputs. However, LN, IN and GN perform normalization on each training data individually, and hence they cannot be used in our low-dimensional situations. Hence, we only examine BN, WN and SELU. BN is applied before activations while for SELU LeCun normal initialization is used (Klambauer et al., 2017). Our simulations show that the neural network can successfully escape from the collapsed areas and approximate the target function with a small error, when BN or SELU are employed. BN changes the weights and biases not only depending on the gradients, and different from ReLU the negative values do not vanish in SELU. However, WN failed because it is only a simple re-parameterization of the weight vectors.

Moreover, our simulations show that the issue of collapse cannot be solved by dropout, which induces sparsity and more zero activations (Srivastava et al., 2014).

## 6 CONCLUSION

We consider here ReLU neural networks for approximating multi-dimensional functions of different regularity, and in particular we focus on deep and narrow NNs due to their reportedly good approximation properties. However, we found that training such NNs is problematic because they converge to erroneous means or partial means or medians of the target function. We demonstrated this collapse problem numerically using one- and two-dimensional functions with $C^0$, $C^\infty$ and $L^2$ regularity. These numerical results are independent of the optimizers we used; the converged state depends on the loss but changing the loss function does not lead to correct answers. In particular, we have observed that the NN with MSE loss converges to the mean or partial mean values while the NN with MAE loss converges to the median values. This collapse phenomenon is induced by the symmetric random initialization, which is popular in practice because it maintains the length of the outputs of each layer as we show theoretically in Section 3.

We analyze theoretically the collapse phenomenon by first proving that if a NN is a constant function then there must exist a layer with output 0 and the gradients of weights and biases in all the previous layers vanish (Lemma 1, Corollary 2, and Lemma 3). Subsequently, we prove that if such conditions are met, then the NN will converge to a constant value depending on the loss function (Theorem 4). Furthermore, if the output of NN is equal to the mean value of the target function, the gradients of weights and biases vanish (Corollaries 5 and 6). In Lemma 7 and Theorem 8 and Proposition 9, we derive estimates of the probability of collapse for general cases, and in Proposition 10, we derive a more precise estimate for deep NNs with width 2. These theoretical estimates are verified numerically by tests using NNs with different layers and widths. Based on these results, we construct a diagram which can be used as a practical guideline in designing deep and narrow NNs that do not suffer from the collapse phenomenon.

Finally, we examine different methods of preventing deep and narrow NNs from converging to erroneous states. In particular, we find that asymmetric initializations including orthogonal initialization and LSUV cannot be used to avoid this collapse. However, some normalization techniques such as batch normalization and SELU can be used successfully to prevent the collapse of deep and narrow NNs; on the other hand, weight normalization fails. Similarly, we examine the effect of dropout which, however, also fails.

ACKNOWLEDGMENTS

This work received support by the DARPA EQUiPS grant N66001-15-2-4055, the NSF grant DMS-1736088, the AFOSR grant FA9550-17-1-0013. The research of the second author was partially supported by the NSF of China 11771083 and the NSF of Fujian 2017J01556, 2016J01013.

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

## A  DERIVATION OF THE LENGTH MAP

For a single input $\mathbf{x}^0$, the normalized squared length of the vector before activation at each layer propagates through the network. Because the weights and biases are independent from $\mathbf{x}^{l-1}$ and are drawn i.i.d. from a zero mean Gaussian with variance $\sigma_w^2/N^{l-1}$ and $\sigma_b^2$, respectively, then

$$\mathbb{E}[\mathbf{W}_{ij}^l \mathbf{x}_j^{l-1}] = \mathbb{E}[\mathbf{W}_{ij}^l]\mathbb{E}[\mathbf{x}_j^{l-1}] = 0,$$

$$\mathrm{Var}(\mathbf{W}_{ij}^l \mathbf{x}_j^{l-1}) = \mathbb{E}\left[(\mathbf{W}_{ij}^l \mathbf{x}_j^{l-1})^2\right] = \mathbb{E}\left[(\mathbf{W}_{ij}^l)^2\right]\mathbb{E}\left[(\mathbf{x}_j^{l-1})^2\right] = \frac{\sigma_w^2}{N^{l-1}}\mathbb{E}\left[(\mathbf{x}_j^{l-1})^2\right]$$

$$= \frac{\sigma_w^2}{N^{l-1}}\mathbb{E}\left[(\phi(\mathbf{h}_j^{l-1}))^2\right],$$

and for $j \neq k$,

$$\mathrm{cov}(\mathbf{W}_{ij}^l \mathbf{x}_j^{l-1}, \mathbf{W}_{ik}^l \mathbf{x}_k^{l-1}) = \mathbb{E}[\mathbf{W}_{ij}^l \mathbf{x}_j^{l-1}\mathbf{W}_{ik}^l \mathbf{x}_k^{l-1}] = 0.$$

We can also see that for a fixed $l \geq 2$, $\mathbf{h}_j^{l-1}$ ($j = 1, 2, \ldots, N^{l-1}$) have the same distribution, and thus $\mathrm{Var}(\mathbf{W}_{ij}^l \mathbf{x}_j^{l-1})$ only depends on $l$. Then $\sum_{j=1}^{N^{l-1}} \mathbf{W}_{ij}^l \mathbf{x}_j^{l-1}$ is a distribution with mean 0 and variance $\sigma_w^2 \mathbb{E}\left[(\phi(\mathbf{h}_1^{l-1}))^2\right]$. Because $\mathbf{h}_i^l = \sum_{j=1}^{N^{l-1}} \mathbf{W}_{ij}^l \mathbf{x}_j^{l-1} + \mathbf{b}_i^l$, then

$$\mathbb{E}[\mathbf{h}_i^l] = 0, \qquad \mathrm{Var}(\mathbf{h}_i^l) = \sigma_w^2 \mathbb{E}\left[(\phi(\mathbf{h}_1^{l-1}))^2\right] + \sigma_b^2.$$

On the other hand,

$$\mathbb{E}[q^l] = \mathbb{E}\left[\frac{1}{N_l}\sum_{i=1}^{N_l}(\mathbf{h}_i^l)^2\right] = \mathbb{E}[(\mathbf{h}_1^l)^2] = \mathrm{Var}(\mathbf{h}_1^l) = \sigma_w^2 \mathbb{E}\left[(\phi(\mathbf{h}_1^{l-1}))^2\right] + \sigma_b^2.$$

Because $\mathbf{h}_1^{l-1} = \sum_{j=1}^{N^{l-2}} \mathbf{W}_{1j}^{l-1}\mathbf{x}_j^{l-2} + \mathbf{b}_1^{l-1}$, if $l = 2$ where $\mathbf{x}_j^0$ is the input, then $\mathbf{h}_1^{l-1}$ is a summation of independent Gaussian random variables and thus is a Gaussian distribution. If $l \geq 3$, by central

limit theorem, $\sum_{j=1}^{N^{l-2}} \mathbf{W}_{1j}^{l-1} \mathbf{x}_j^{l-2}$ converges in distribution to a Gaussian distribution as $N^{l-2} \to \infty$, so is $\mathbf{h}_1^{l-1}$. Using $\mathrm{Var}(\mathbf{h}_1^{l-1}) = \mathbb{E}[q^{l-1}]$, we have

$$\mathbb{E}\left[(\phi(\mathbf{h}_1^{l-1}))^2\right] = \mathbb{E}\left[\left(\phi\left(\sqrt{\mathbb{E}[q^{l-1}]}\frac{\mathbf{h}_1^{l-1}}{\sqrt{\mathbb{E}[q^{l-1}]}}\right)\right)^2\right] \to \int \phi(\sqrt{\mathbb{E}[q^{l-1}]}z)^2 \mathcal{D}z, \quad \text{as } N^{l-2} \to \infty,$$

where $\mathcal{D}z = \frac{dz}{\sqrt{2\pi}}e^{-\frac{z^2}{2}}$ is the standard Gaussian measure. Therefore,

$$\mathbb{E}[q^2] = \sigma_w^2 \int \mathcal{D}z\phi(\sqrt{\mathbb{E}[q^1]}z)^2 + \sigma_b^2,$$

$$\mathbb{E}[q^l] \to \sigma_w^2 \int \mathcal{D}z\phi(\sqrt{\mathbb{E}[q^{l-1}]}z)^2 + \sigma_b^2, \quad \text{as } N^{l-2} \to \infty \quad \text{for } l \geq 3.$$

## B   PROOF OF LEMMA 1

**Lemma 11.** *Let $\mathbf{A} \in \mathbb{R}^{n \times m}$ be a random matrix, where $\{\mathbf{A}_{ij}\}_{i \in \{1,2,\dots,n\}, j \in \{1,2,\dots,m\}}$ are random variables, and the joint distribution of $(\mathbf{A}_{i1}, \mathbf{A}_{i2}, \dots, \mathbf{A}_{im})$ is absolutely continuous for $i = 1, 2, \dots, n$. If $\mathbf{x} \in \mathbb{R}^m$ is a nonzero column vector, then $\mathbb{P}(\mathbf{A}\mathbf{x} = \mathbf{0}) = 0$.*

*Proof.* Let us consider the first value of $\mathbf{A}\mathbf{x}$, i.e., $\sum_{j=1}^m \mathbf{A}_{1j}\mathbf{x}_j$. Because $\mathbf{x} \neq \mathbf{0}$, we have $\{\sum_{j=1}^m \mathbf{A}_{1j}\mathbf{x}_j = 0\}$ is a hyperplane in $\mathbb{R}^m$ whose coordinates are $\mathbf{A}_{1j}$, $j = 1, 2, \dots, m$. Because the joint distribution of $(\mathbf{A}_{11}, \mathbf{A}_{12}, \dots, \mathbf{A}_{1m})$ is absolutely continuous, $\mathbb{P}(\sum_{j=1}^m \mathbf{A}_{1j}\mathbf{x}_j = 0) = 0$. Hence,

$$0 \leq \mathbb{P}(\mathbf{A}\mathbf{x} = \mathbf{0}) = \mathbb{P}(\sum_{j=1}^m \mathbf{A}_{ij}\mathbf{x}_j = 0 \quad \forall i = 1, 2, \dots, n) \leq \mathbb{P}(\sum_{j=1}^m \mathbf{A}_{1j}\mathbf{x}_j = 0) = 0.$$

Therefore, $\mathbb{P}(\mathbf{A}\mathbf{x} = \mathbf{0}) = 0$. $\qquad\square$

Now let us go back to the proof of Lemma 1.

*Proof.* By assumption A2 and Lemma 11, $\mathcal{N}(\mathbf{x}^0) = \mathbf{W}^L\mathbf{x}^{L-1}(\mathbf{x}^0) + \mathbf{b}^L$ is a constant function, iff $\mathbf{x}^{L-1}(\mathbf{x}^0)$ is a constant function with respect to $\mathbf{x}^0$. So we can assume that there is ReLU in the last layer, and prove that there exists a layer $l \in \{1, \dots, L\}$, s.t., $\mathbf{h}^l \leq \mathbf{0}$ and $\mathbf{x}^l = \mathbf{0}$ wp1 for every $\mathbf{x}^0 \in \Omega$. We proceed in two steps.

i) For $L = 1$, we have $\mathbf{x}^1 = \mathrm{ReLU}(\mathbf{h}) = \mathrm{ReLU}(\mathbf{W}\mathbf{x}^0 + \mathbf{b})$ is a constant. If $\mathbf{h}$ is not always $\leq \mathbf{0}$, then there exists $\tilde{\mathbf{x}}^0 \in \Omega$ and $k$, s.t., $\mathbf{h}_k(\tilde{\mathbf{x}}^0) > 0$. Because $\Omega \subset \mathbb{R}^{d_{in}}$ is a connected space with at least two points, then $\Omega$ has no isolated points, which implies $\tilde{\mathbf{x}}^0$ is not an isolated point. Since the neural network is a continuous map, $\Omega^1 = \{\mathbf{x}^1(\mathbf{x}^0) : \mathbf{x}^0 \in \Omega\}$ is connected. So there exists $\hat{\mathbf{x}}^0 \neq \tilde{\mathbf{x}}^0$ in the neighborhood of $\tilde{\mathbf{x}}^0$, s.t., $\mathbf{h}_k(\hat{\mathbf{x}}^0) > 0$ and $\mathbf{h}_k(\hat{\mathbf{x}}^0) \neq \mathbf{h}_k(\tilde{\mathbf{x}}^0)$ wp1, because of $\mathbb{P}(\mathbf{W}(\hat{\mathbf{x}}^0 - \tilde{\mathbf{x}}^0) = \mathbf{0}) = 0$ by Lemma 11. Hence, $\mathbf{x}^1(\tilde{\mathbf{x}}^0) \neq \mathbf{x}^1(\hat{\mathbf{x}}^0)$, which contradicts the fact that $\mathbf{x}^1$ is a constant function. Therefore, $\mathbf{h} \leq \mathbf{0}$ and $\mathbf{x}^1 = \mathbf{0}$.

ii) Assume the theorem is true for $L$. Then for $L + 1$, if $\mathbf{x}^1 = 0$, choose $l = 1$ and we are done; otherwise, consider the NN without the first layer with $\mathbf{x}^1 \in \Omega^1$ as the input, denoted $\mathcal{N}_1$. By i, $\Omega^1$ is a connected space with at least two points. Because $\mathcal{N}_1$ is a constant function of $\mathbf{x}^1$ and has $L$ layers, by induction, there exists a layer whose output is zero. Therefore, for the original neural network $\mathcal{N}$, the output of such layer is also zero.

By i and ii, the statement is true for any $L$. $\qquad\square$

## C  PROOF OF COROLLARY 2

*Proof.* By Lemma 1, there exists a layer $l \in \{1, \ldots, L-1\}$, s.t. $\mathbf{h}^l \leq \mathbf{0}$ and $\mathbf{x}^l = \mathbf{0}$ wp1. Because $\mathcal{N}$ is bias-free, $\mathbf{h}^{l+1} = \mathbf{W}^{l+1}\mathbf{x}^l = \mathbf{0}$ and $\mathbf{x}^{l+1} = \text{ReLU}(\mathbf{h}^{l+1}) = \mathbf{0}$ wp1. By induction, for any $n \geq l$, $\mathbf{h}^n \leq \mathbf{0}$ and $\mathbf{x}^n = \mathbf{0}$ wp1. □

## D  PROOF OF LEMMA 3

*Proof.* Because $\mathbf{x}^l \equiv \mathbf{0}$, it is then obvious by backpropagation. □

## E  PROOF OF THEOREM 4

*Proof.* Because $\mathbf{x}^l(\mathbf{x}^0) \equiv \mathbf{0}$, $\mathcal{N}(\mathbf{x}^0)$ is a constant function, and then by Lemma 3, gradients of the loss function w.r.t. the weights and biases in layers $1, \ldots, l$ vanish. Hence, the weights and biases in layers $1, \ldots, l$ will not change when using a gradient based optimizer, which implies $\mathcal{N}(\mathbf{x}^0)$ is always a constant function depending on the weights and biases in layers $l+1, \ldots, L$. Therefore, $\mathcal{N}$ will be optimized to a constant function, which has the smallest loss. For $L^2$ loss, this constant with the smallest loss is $\mathbb{E}[\mathbf{y}]$. For $L^1$ loss, this constant with the smallest loss is its median. □

## F  PROOF OF COROLLARY 5

*Proof.* Because $\mathcal{N}(\mathbf{x}^0)$ is a constant function, by Lemma 1 and Theorem 4, $\mathcal{N}$ is optimized to $\mathbb{E}[\mathbf{y}]$. Also, since $\mathcal{N}$ is equal to $\mathbb{E}[\mathbf{y}]$, gradients vanish. □

## G  PROOF OF COROLLARY 6

*Proof.* It suffices to show that gradients vanish for $\mathbf{x}^0 \in K_i$, $i = 1, \ldots, n$ and $\mathbf{x}^0 \in \Omega \setminus \cup_{i=1}^n K_i$.

i) When $\mathbf{x}^0$ is restricted on $K_i$, $\mathcal{N}(\mathbf{x}^0)$ is a constant function with value $\mathbb{E}_{\mathbf{x}_{K_i}^0}[\mathbf{y}(\mathbf{x}_{K_i}^0)]$. Similar to Corollary 5, gradients vanish when using the $L^2$ loss.

ii) For $\mathbf{x}^0 \in \Omega \setminus \cup_{i=1}^n K_i$, the loss at $\mathbf{x}^0$ is 0, so gradients vanish.

By i and ii, gradients vanish when using the $L^2$ (MSE) loss. □

## H  PROOF OF LEMMA 7

*Proof.* Let $\mathbf{x} = (x_1, x_2, \ldots, x_{d_{in}})$ be any input, and $\mathbf{y} = (y_1, y_2, \ldots, y_{d_{out}})$ be the corresponding output. For $i = 1, \ldots, d_{out}$,

$$y_i = \text{ReLU}(\mathbf{w}_i \cdot \mathbf{x} + b_i) = \text{ReLU}((w_{i1}, \ldots, w_{id_{in}}, b_i) \cdot (x_1, x_2, \ldots, x_{d_{in}}, 1)).$$

Because $(w_{i1}, \ldots, w_{id_{in}}, b_i)$ is a $(d_{in} + 1)$-dim vector initialized by a symmetric distribution, then

$$\mathbb{P}((w_{i1}, \ldots, w_{id_{in}}, b_i) \cdot (x_1, x_2, \ldots, x_{d_{in}}, 1) > 0) = \frac{1}{2}.$$

So $\mathbb{P}(y_i = 0) = \frac{1}{2}$, and then $\mathbb{P}(\mathbf{y} = \mathbf{0}) = \Pi_{i=1}^{d_{out}}\mathbb{P}(y_i = 0) = (\frac{1}{2})^{d_{out}}$. Here $\mathbb{P}$ denotes the probability. □

## I  PROOF OF THEOREM 8

*Proof.* If the last layer also employs ReLU activation, by Lemma 7, $\mathbb{P}(\mathbf{x}^l = \mathbf{0}|\mathbf{x}^{l-1} \neq \mathbf{0}) = (1/2)^{N^l}$ for $l = 1, \ldots, L$. Then, for any fixed input $\mathbf{x}^0 \neq \mathbf{0}$,

$$\mathbb{P}(\mathbf{x}^L \neq \mathbf{0}) = \mathbb{P}(\mathbf{x}^{L-1} \neq \mathbf{0})\mathbb{P}(\mathbf{x}^L \neq \mathbf{0}|\mathbf{x}^{L-1} \neq \mathbf{0}) + \mathbb{P}(\mathbf{x}^{L-1} = \mathbf{0})\mathbb{P}(\mathbf{x}^L \neq \mathbf{0}|\mathbf{x}^{L-1} = \mathbf{0})$$

$$= \mathbb{P}(\mathbf{x}^{L-1} \neq \mathbf{0})\mathbb{P}(\mathbf{x}^L \neq \mathbf{0}|\mathbf{x}^{L-1} \neq \mathbf{0}) = \mathbb{P}(\mathbf{x}^{L-1} \neq \mathbf{0})(1 - (1/2)^{N^l}) = \cdots = \Pi_{l=1}^L(1 - (1/2)^{N^l}).$$

The last equality holds because $\mathbb{P}(\mathbf{x}^0 \neq \mathbf{0}) = 1$.

If in the last layer we do not apply ReLU activation, then $\mathbb{P}(\mathbf{x}^L \neq \mathbf{0}) = \mathbb{P}(\mathbf{x}^{L-1} \neq \mathbf{0}) = \Pi_{l=1}^{L-1}(1 - (1/2)^{N^l})$. $\qquad \square$

## J  PROOF OF PROPOSITION 9

*Proof.* If the last layer also has ReLU activation, by Lemma 7,

$$\mathbb{P}(\mathbf{x}^L = \mathbf{0}) = \mathbb{P}(\mathbf{x}^{L-1} \neq \mathbf{0})\mathbb{P}(\mathbf{x}^L = \mathbf{0}|\mathbf{x}^{L-1} \neq \mathbf{0}) + \mathbb{P}(\mathbf{x}^{L-1} = \mathbf{0})\mathbb{P}(\mathbf{x}^L = \mathbf{0}|\mathbf{x}^{L-1} = \mathbf{0})$$
$$= \mathbb{P}(\mathbf{x}^{L-1} \neq \mathbf{0})(1/2)^{N^L} + \mathbb{P}(\mathbf{x}^{L-1} = \mathbf{0})(1/2)^{N^L} = (1/2)^{N^L}.$$

If the last layer does not have ReLU activation, and $L \geq 2$, then

$$\mathbb{P}(\mathbf{x}^L = \mathbf{b}_n) = \mathbb{P}(\mathbf{x}^{L-1} = \mathbf{0}) = (1/2)^{N^{L-1}}.$$

For $L = 1$, $\mathcal{N}$ is a single layer perceptron, which is a trivial case. $\qquad \square$

## K  PROOF OF PROPOSITION 10

*Proof.* We consider a ReLU neural network with $d_{in} = 1$ and each hidden layer with width 2. Because all biases are zero, then it is easy to see the following fact: when the input is 0, the output of any neuron in any layer is 0; when the input is negative, the output of any neuron in any layer is a linear function with respect to the input; when the input is positive, the output of any neuron in any layer is also a linear function with respect to the input. Because the origin is an interior point of $\Omega$, then it suffices to consider a subset $[-a, a] \subset \Omega$ with $a \in \mathbb{R}^+$. The output of each hidden layer has 16 possible cases:

$$\text{case (1):} \begin{cases} \begin{pmatrix} \omega_1 x \\ \omega_2 x \end{pmatrix}, & x \in [0, a] \\ \begin{pmatrix} \omega_1^* x \\ \omega_2^* x \end{pmatrix}, & x \in [-a, 0] \end{cases}, \quad \text{case (2):} \begin{cases} \begin{pmatrix} \omega_1 x \\ \omega_2 x \end{pmatrix}, & x \in [0, a] \\ \begin{pmatrix} \omega_1^* x \\ 0 \end{pmatrix}, & x \in [-a, 0] \end{cases},$$

$$\text{case (3):} \begin{cases} \begin{pmatrix} \omega_1 x \\ \omega_2 x \end{pmatrix}, & x \in [0, a] \\ \begin{pmatrix} 0 \\ \omega_2^* x \end{pmatrix}, & x \in [-a, 0] \end{cases}, \quad \text{case (4):} \begin{cases} \begin{pmatrix} \omega_1 x \\ \omega_2 x \end{pmatrix}, & x \in [0, a] \\ \begin{pmatrix} 0 \\ 0 \end{pmatrix}, & x \in [-a, 0] \end{cases},$$

$$\text{case (5):} \begin{cases} \begin{pmatrix} \omega_1 x \\ 0 \end{pmatrix}, & x \in [0, a] \\ \begin{pmatrix} \omega_1^* x \\ \omega_2^* x \end{pmatrix}, & x \in [-a, 0] \end{cases}, \quad \text{case (6):} \begin{cases} \begin{pmatrix} \omega_1 x \\ 0 \end{pmatrix}, & x \in [0, a] \\ \begin{pmatrix} \omega_1^* x \\ 0 \end{pmatrix}, & x \in [-a, 0] \end{cases},$$

$$\text{case (7):} \begin{cases} \begin{pmatrix} \omega_1 x \\ 0 \end{pmatrix}, & x \in [0, a] \\ \begin{pmatrix} 0 \\ \omega_2^* x \end{pmatrix}, & x \in [-a, 0] \end{cases}, \quad \text{case (8):} \begin{cases} \begin{pmatrix} \omega_1 x \\ 0 \end{pmatrix}, & x \in [0, a] \\ \begin{pmatrix} 0 \\ 0 \end{pmatrix}, & x \in [-a, 0] \end{cases},$$

$$\text{case (9):} \begin{cases} \begin{pmatrix} 0 \\ \omega_2 x \end{pmatrix}, & x \in [0, a] \\ \begin{pmatrix} \omega_1^* x \\ \omega_2^* x \end{pmatrix}, & x \in [-a, 0] \end{cases}, \quad \text{case (10):} \begin{cases} \begin{pmatrix} 0 \\ \omega_2 x \end{pmatrix}, & x \in [0, a] \\ \begin{pmatrix} \omega_1^* x \\ 0 \end{pmatrix}, & x \in [-a, 0] \end{cases},$$

$$\text{case (11): } \begin{cases} \begin{pmatrix} 0 \\ \omega_2 x \end{pmatrix}, & x \in [0, a] \\ \begin{pmatrix} 0 \\ \omega_2^* x \end{pmatrix}, & x \in [-a, 0] \end{cases}, \quad \text{case (12): } \begin{cases} \begin{pmatrix} 0 \\ \omega_2 x \end{pmatrix}, & x \in [0, a] \\ \begin{pmatrix} 0 \\ 0 \end{pmatrix}, & x \in [-a, 0] \end{cases},$$

$$\text{case (13): } \begin{cases} \begin{pmatrix} 0 \\ 0 \end{pmatrix}, & x \in [0, a] \\ \begin{pmatrix} \omega_1^* x \\ \omega_2^* x \end{pmatrix}, & x \in [-a, 0] \end{cases}, \quad \text{case (14): } \begin{cases} \begin{pmatrix} 0 \\ 0 \end{pmatrix}, & x \in [0, a] \\ \begin{pmatrix} \omega_1^* x \\ 0 \end{pmatrix}, & x \in [-a, 0] \end{cases},$$

$$\text{case (15): } \begin{cases} \begin{pmatrix} 0 \\ 0 \end{pmatrix}, & x \in [0, a] \\ \begin{pmatrix} 0 \\ \omega_2^* x \end{pmatrix}, & x \in [-a, 0] \end{cases}, \quad \text{case (16): } \begin{cases} \begin{pmatrix} 0 \\ 0 \end{pmatrix}, & x \in [0, a] \\ \begin{pmatrix} 0 \\ 0 \end{pmatrix}, & x \in [-a, 0] \end{cases},$$

where $w_1, w_2, w_1^*, w_2^*$ are some coefficients.

Each case in the $l$th hidden layer may also induce all 16 cases in the $(l+1)$th layer. For any given case in the $l$th hidden layer, we will compute the probabilities of these 16 cases for the $(l+1)$th layer as follows.

i) Case (1)

Note that $\boldsymbol{\omega} = (\omega_1, \omega_2)$ lies in the first quadrant, and $\boldsymbol{\omega}^* = (\omega_1^*, \omega_2^*)$ lies in the third quadrant. Then the output of the next layer is

$$\begin{cases} \begin{pmatrix} \text{ReLU}((A_{11}\omega_1 + A_{12}\omega_2)x) \\ \text{ReLU}((A_{21}\omega_1 + A_{22}\omega_2)x) \end{pmatrix}, & x \in [0, a] \\ \begin{pmatrix} \text{ReLU}((A_{11}\omega_1^* + A_{12}\omega_2^*)x) \\ \text{ReLU}((A_{21}\omega_1^* + A_{22}\omega_2^*)x) \end{pmatrix}, & x \in [-a, 0] \end{cases}.$$

Since the matrix $\begin{pmatrix} A_{11} & A_{12} \\ A_{21} & A_{22} \end{pmatrix}$ is random, for fixed $\boldsymbol{\omega}$ and $\boldsymbol{\omega}^*$, the probability of case (1) is $\left(\frac{\angle(\boldsymbol{\omega}, \boldsymbol{\omega}^*)}{2\pi}\right)^2$. Without loss of generality, we can assume that $\|\boldsymbol{\omega}\| = \|\boldsymbol{\omega}^*\| = 1$, and hence we can assume that $\boldsymbol{\omega} = (\cos\theta, \sin\theta)$, $\theta \in (0, \frac{\pi}{2})$ and $\boldsymbol{\omega}^* = (\cos\psi, \sin\psi)$, $\psi \in (\pi, \frac{3\pi}{2})$. It is easy to see that

$$\angle(\boldsymbol{\omega}, \boldsymbol{\omega}^*) = \begin{cases} \psi - \theta, & \psi \leq \theta + \pi \\ 2\pi + \theta - \psi, & \psi > \theta + \pi \end{cases}.$$

Since $\boldsymbol{\omega}, \boldsymbol{\omega}^*$ are random, the probability of case (1) is

$$\frac{2^2}{\pi^2} \int_0^{\frac{\pi}{2}} d\theta \int_\pi^{\frac{3}{2}\pi} \left(\frac{\angle(\boldsymbol{\omega}, \boldsymbol{\omega}^*)}{2\pi}\right)^2 d\psi = \frac{17}{96}.$$

Similarly, the probability of cases (6), (11) and (16) in the $(l+1)$th layer are also $\frac{17}{96}$. For cases (2), (3), (5), (8), (9), (12), (14) and (15), the probability is

$$\frac{2^2}{\pi^2} \int_0^{\frac{\pi}{2}} d\theta \int_\pi^{\frac{3}{2}\pi} \frac{\angle(\boldsymbol{\omega}, \boldsymbol{\omega}^*)}{2\pi} \cdot \frac{2\pi - \angle(\boldsymbol{\omega}, \boldsymbol{\omega}^*)}{2\pi} d\psi = \frac{1}{32}.$$

For cases (4), (7), (10) and (13), the probability is

$$\frac{2^2}{\pi^2} \int_0^{\frac{\pi}{2}} d\theta \int_\pi^{\frac{3}{2}\pi} \left(\frac{2\pi - \angle(\boldsymbol{\omega}, \boldsymbol{\omega}^*)}{2\pi}\right)^2 d\psi = \frac{1}{96}.$$

ii) Case (2) (the same method can be applied for cases (3), (5) and (9))

Note that in this case we can assume that $\boldsymbol{\omega} = (\cos\theta, \sin\theta)$, $\theta \in (0, \frac{\pi}{2})$ and $\boldsymbol{\omega}^* = (-1, 0)$ is a constant vector. It is easy to see that $\angle(\boldsymbol{\omega}, \boldsymbol{\omega}^*) = \pi - \theta$, and hence the probabilities of cases (1), (6), (11) and (16) are

$$\frac{2}{\pi} \int_0^{\frac{\pi}{2}} \left(\frac{\angle(\boldsymbol{\omega}, \boldsymbol{\omega}^*)}{2\pi}\right)^2 d\theta = \frac{7}{48}.$$

Similarly, the probabilities of cases (2), (3), (5), (8), (9), (12), (14) and (15) are

$$\frac{2}{\pi} \int_0^{\frac{\pi}{2}} \frac{\angle(\boldsymbol{\omega}, \boldsymbol{\omega}^*)}{2\pi} \cdot \frac{2\pi - \angle(\boldsymbol{\omega}, \boldsymbol{\omega}^*)}{2\pi} d\theta = \frac{1}{24},$$

and the probabilities of cases (4), (7), (10) and (13) are

$$\frac{2}{\pi} \int_0^{\frac{\pi}{2}} \left(\frac{2\pi - \angle(\boldsymbol{\omega}, \boldsymbol{\omega}^*)}{2\pi}\right)^2 d\theta = \frac{1}{48}.$$

iii) Case (4) (the same method can be applied for cases (8) and (12))

The output of the next layer is

$$\begin{cases} \begin{pmatrix} \text{ReLU}((A_{11}\omega_1 + A_{12}\omega_2)x) \\ \text{ReLU}((A_{21}\omega_1 + A_{22}\omega_2)x) \end{pmatrix}, & x \in [0, a] \\ \begin{pmatrix} 0 \\ 0 \end{pmatrix}, & x \in [-a, 0] \end{cases}.$$

It is easy to see that the probabilities of cases (4), (8), (12) and (16) are $\frac{1}{4}$, and the probabilities of all other cases are 0.

iv) Case (6) (the same method can be applied for case (11))

The output of the next layer is

$$\begin{cases} \begin{pmatrix} \text{ReLU}(A_{11}\omega_1 x) \\ \text{ReLU}(A_{21}\omega_1 x) \end{pmatrix}, & x \in [0, a] \\ \begin{pmatrix} \text{ReLU}(A_{11}\omega_1^* x) \\ \text{ReLU}(A_{21}\omega_1^* x) \end{pmatrix}, & x \in [-a, 0] \end{cases}.$$

Note that in this case, $\omega_1 > 0$ and $\omega_1^* < 0$, and thus it is not hard to see that the probabilities of cases (1), (6), (11) and (16) are $\frac{1}{4}$, and the probabilities of all the other cases are 0.

v) Case (7) (the same method can be applied for case (10))

The output of the next layer is

$$\begin{cases} \begin{pmatrix} \text{ReLU}(A_{11}\omega_1 x) \\ \text{ReLU}(A_{21}\omega_1 x) \end{pmatrix}, & x \in [0, a] \\ \begin{pmatrix} \text{ReLU}(A_{12}\omega_2^* x) \\ \text{ReLU}(A_{22}\omega_2^* x) \end{pmatrix}, & x \in [-a, 0] \end{cases}.$$

Therefore, the probabilities of all the 16 cases are $\frac{1}{16}$.

vi) Case (13) (the same method can be applied for cases (14) and (15))

Similar to the argument of the case (4), it is easily to see that the probabilities for cases (13), (14), (15) and (16) are $\frac{1}{4}$, and the probabilities for all other cases are 0.

vii) Case (16)

The output of the next layer is the case (16) with probability 1.

By i, ii, iii, iv, v, vi and vii, we can get the probability transition matrix

$$P = \begin{bmatrix}
\frac{17}{96} & \frac{7}{48} & \frac{7}{48} & 0 & \frac{7}{48} & \frac{1}{4} & \frac{1}{16} & 0 & \frac{7}{48} & \frac{1}{16} & \frac{1}{4} & 0 & 0 & 0 & 0 & 0 \\
\frac{1}{32} & \frac{1}{24} & \frac{1}{24} & 0 & \frac{1}{24} & 0 & \frac{1}{16} & 0 & \frac{1}{24} & \frac{1}{16} & 0 & 0 & 0 & 0 & 0 & 0 \\
\frac{1}{32} & \frac{1}{24} & \frac{1}{24} & 0 & \frac{1}{24} & 0 & \frac{1}{16} & 0 & \frac{1}{24} & \frac{1}{16} & 0 & 0 & 0 & 0 & 0 & 0 \\
\frac{1}{96} & \frac{1}{48} & \frac{1}{48} & \frac{1}{4} & \frac{1}{48} & 0 & \frac{1}{16} & \frac{1}{4} & \frac{1}{48} & \frac{1}{16} & 0 & \frac{1}{4} & 0 & 0 & 0 & 0 \\
\frac{1}{32} & \frac{1}{24} & \frac{1}{24} & 0 & \frac{1}{24} & 0 & \frac{1}{16} & 0 & \frac{1}{24} & \frac{1}{16} & 0 & 0 & 0 & 0 & 0 & 0 \\
\frac{17}{96} & \frac{7}{48} & \frac{7}{48} & 0 & \frac{7}{48} & \frac{1}{4} & \frac{1}{16} & 0 & \frac{7}{48} & \frac{1}{16} & \frac{1}{4} & 0 & 0 & 0 & 0 & 0 \\
\frac{1}{96} & \frac{1}{48} & \frac{1}{48} & 0 & \frac{1}{48} & 0 & \frac{1}{16} & 0 & \frac{1}{48} & \frac{1}{16} & 0 & 0 & 0 & 0 & 0 & 0 \\
\frac{1}{32} & \frac{1}{24} & \frac{1}{24} & \frac{1}{4} & \frac{1}{24} & 0 & \frac{1}{16} & \frac{1}{4} & \frac{1}{24} & \frac{1}{16} & 0 & \frac{1}{4} & 0 & 0 & 0 & 0 \\
\frac{1}{32} & \frac{1}{24} & \frac{1}{24} & 0 & \frac{1}{24} & 0 & \frac{1}{16} & 0 & \frac{1}{24} & \frac{1}{16} & 0 & 0 & 0 & 0 & 0 & 0 \\
\frac{1}{96} & \frac{1}{48} & \frac{1}{48} & 0 & \frac{1}{48} & 0 & \frac{1}{16} & 0 & \frac{1}{48} & \frac{1}{16} & 0 & 0 & 0 & 0 & 0 & 0 \\
\frac{17}{96} & \frac{7}{48} & \frac{7}{48} & 0 & \frac{7}{48} & \frac{1}{4} & \frac{1}{16} & 0 & \frac{7}{48} & \frac{1}{16} & \frac{1}{4} & 0 & 0 & 0 & 0 & 0 \\
\frac{1}{32} & \frac{1}{24} & \frac{1}{24} & \frac{1}{4} & \frac{1}{24} & 0 & \frac{1}{16} & \frac{1}{4} & \frac{1}{24} & \frac{1}{16} & 0 & \frac{1}{4} & 0 & 0 & 0 & 0 \\
\frac{1}{96} & \frac{1}{48} & \frac{1}{48} & 0 & \frac{1}{48} & 0 & \frac{1}{16} & 0 & \frac{1}{48} & \frac{1}{16} & 0 & 0 & \frac{1}{4} & \frac{1}{4} & \frac{1}{4} & 0 \\
\frac{1}{32} & \frac{1}{24} & \frac{1}{24} & 0 & \frac{1}{24} & 0 & \frac{1}{16} & 0 & \frac{1}{24} & \frac{1}{16} & 0 & 0 & \frac{1}{4} & \frac{1}{4} & \frac{1}{4} & 0 \\
\frac{1}{32} & \frac{1}{24} & \frac{1}{24} & 0 & \frac{1}{24} & 0 & \frac{1}{16} & 0 & \frac{1}{24} & \frac{1}{16} & 0 & 0 & \frac{1}{4} & \frac{1}{4} & \frac{1}{4} & 0 \\
\frac{17}{96} & \frac{7}{48} & \frac{7}{48} & \frac{1}{4} & \frac{7}{48} & \frac{1}{4} & \frac{1}{16} & \frac{1}{4} & \frac{7}{48} & \frac{1}{16} & \frac{1}{4} & \frac{1}{4} & \frac{1}{4} & \frac{1}{4} & \frac{1}{4} & 1
\end{bmatrix},$$

where $P_{ji}$ is the probability of that the $(l+1)$th layer is case $j$ when the $i$th layer is case $i$.

Furthermore, direct computations show that the probability distribution vector of the first hidden layer $\pi^1$ is

$$\left(0, 0, 0, \frac{1}{4}, 0, 0, \frac{1}{4}, 0, 0, \frac{1}{4}, 0, 0, \frac{1}{4}, 0, 0, 0\right)^T.$$

Therefore, the probability distribution of the $l$th hidden layer is

$$\pi^l = P^{l-1}\pi^1.$$

$\square$

