# OpenReview forum: "Collapse of deep and narrow neural nets"
_ICLR.cc/2019/Conference_

### Official Review · AnonReviewer1 · 2018-10-31
**Good, interesting results - presentation improved**

**Rating:** 7
**Confidence:** 5

**Review:**

The paper studies failure modes of deep and narrow networks. I find this research extremely valuable and interesting. In addition to that, the paper focuses on as small as possible models, for which the undesired behavior occurs. That is another great positive, too much of a research in DL focuses on the most complex and general cases in my opinion. I would be more than happy to give this paper a very strong recommendation, if not for numerous flaws in presentation. If those get improved, I am very eager to increase my rating. Here are the things that I think need an improvement:
1. The formulation of theorems.
The paper strives for mathematical style. Yet the formulations of the theorems are very colloquial. Expression "by assuming random weights" is not what one wants to see in a rigorous math paper. The formulations of the theorems need to be made rigorous and easy to understand, the assumptions need to be clearly stated and all concepts used strictly defined.
2. Too many theorems
9 (!) theorems is way too much. Theorem is a significant contribution. I strongly suggest having 1-2 strong theorems, and downgrading more technical lemmas to a lemma and proposition status.
In addition - the problem studied is really a study of bad local minimas for neural networks. More mentions of the previous work related to the topic would improve the scientific quality additionally, in my opinion.

---

> ### Author Response · Authors · 2018-11-26
> **Response to Reviewer 1**
>
> We would like to thank reviewer 1 whose constructive comments helped us improve our manuscript. We have totally rewritten our theorems to follow a more mathematical exposition. Below, we provide point-to-point answers to the reviewer’s concerns and the edits are marked in blue in the revised manuscript.
>
> 1) The formulation of theorems. The paper strives for mathematical style. Yet the formulations of the theorems are very colloquial. Expression "by assuming random weights" is not what one wants to see in a rigorous math paper. The formulations of the theorems need to be made rigorous and easy to understand, the assumptions need to be clearly stated and all concepts used strictly defined.
>
> Answer: We agree that the exposition of the theorems can be improved so we have tried our best to improve the manuscript. All theorems have been revised based on the reviewer’s comments. Please see more details highlighted in the blue color on pages 5, 6 and 7 of the revised manuscript.
>
> 2) Too many theorems 9 (!) theorems is way too much. Theorem is a significant contribution. I strongly suggest having 1-2 strong theorems, and downgrading more technical lemmas to a lemma and proposition status.
>
> Answer:  We thank the reviewer for pointing out the inaccurate usage of theorems. We have downgraded most theorems to lemmas, corollaries and propositions with only Theorem 4 and Theorem 8 as the main theorems.
>
> 3) In addition - the problem studied is really a study of bad local minima for neural networks. More mentions of the previous work related to the topic would improve the scientific quality additionally, in my opinion.
>
> Answer: We thank the reviewer for pointing out this. We added an introduction to the work on local minima and cited several papers on this topic. Please see more details highlighted in the blue color on page 2 of the revised manuscript.

---

### Official Review · AnonReviewer3 · 2018-11-01
**Some interesting results, but not good enough.**

**Rating:** 4
**Confidence:** 4

**Review:**

In this paper, the authors investigate the collapse of deep and shallow network to a constant function under the following setting:
1. Very shallow networks, width ~ 10
2. ReLU activation function.
3. Symmetric weights and biases initialization.
4. Vanilla feed forward networks.

The main take-home message is: don't use neural networks (NNs) that are both deep and shallow.

The theoretical analysis is built on the observation:
1. Every neuron (after applying ReLU) is equal to zero with probability 1/2.
2. For narrow network and for any fixed input, there is a high chance that all neurons in a particular hidden layer are all zero. All neurons after that layer are all zero if zero-bias initialization is used.
3. The authors conclude that derivatives of all parameters vanish but the bias of the last layer.
4. As a result, the network collapse to its mean (median) if mean squared loss (L1 loss) is used because only the bias of the last layer is being updated during training.

Pros.
1. I think the phenomenon that shallow and deep NNs collapse to a constant is very interesting.
2. The authors provide empirical and theoretical insights in favor of wider networks: have a better chance to avoid vanishing gradients.
3. For shallow networks, it might be better not to use ReLU.

Cons:
1.The analysis works in a very limited setting, works for ReLU but not other activations: tanh, erf, SELU etc.
2. Very shallow networks are not popular in practice. Width>=100 is popular for fully-connected layers.
3. The phenomenon observed by the paper can easily be addressed using any of the following trick:
   3.1. Non-symmetric initialization  (set the mean to be non-zero).
   3.2 . wider networks.
4. Most of the analysis (and theorems)  are about one single input. In another word, distribution of the inputs have not been taken into account.
5. I don't think the author provides a completely rigorous justification for the collapse phenomenon.

Other comments.
1. Eq (2) in page 4 is not trivially correct. The expectation operator (w.r.p. to lower layers) is moved into the activation function phi, justification is needed for this step.
2. Theorem 4: when the Lebesgue measure of $\Omega$ is NOT finite, it is unclear how to define a uniform probability distribution on it.
3. Theorem 4: the integrability assumption on y should depend on the loss: for L2 loss (L1 loss), squared (absolutely) integrable  should be used. They are not the same.

---

> ### Author Response · Authors · 2018-11-26
> **Response to Reviewer 3 (Part II)**
>
> 3) The phenomenon observed by the paper can easily be addressed using any of the following trick: 3.1. Non-symmetric initialization (set the mean to be non-zero). 3.2. wider networks.
>
> Answer: Thanks for your comment. We agree that the collapse phenomenon can be addressed by wider networks. However, it will increase the computational cost. Non-symmetric initialization can reduce the probability of collapse but not eliminated it totally. By contrast, our paper provides a priori knowledge as to what is the minimum width required to avoid collapse with specific probability. This is like having a phase diagram in thermodynamics, which we use only when we need it to design a new material! Moreover, this is the first time to analyze this phenomenon both theoretically and numerically, and our analysis helps us understand this quantitatively. The non-symmetric initialization proposed in the literature mainly includes orthogonal initialization and layer-sequential unit-variance initialization, but they cannot be used to avoid the collapse, as we have discussed in Section 5.1. Using wider networks can reduce the probability of collapse, and we provide a quantitative guideline of how wider should be chosen. In our opinion, this is new fundamental knowledge that merits publication.
>
> 4) Most of the analysis (and theorems) are about one single input. In another word, distribution of the inputs have not been taken into account.
>
> Answer:  We apologize for the confusion of the statements in theorems. Lemma 1, Corollary 2, Lemma 3 require that the network is a constant function, and thus does not depend on the distribution of the input. Theorem 4 and Corollaries 5 and 6 depend on the input distribution. In our original submission, we chose the uniform distribution as an example, and in fact these results can be generalized to other distributions. The probability calculated in Proposition 10 only requires that the origin is an interior point of the input domain. Lemma 7, Theorem 8 and Proposition 9 consider one single input, because these results are the estimation of the probability of NN to collapse. Please see our revised statements highlighted in the blue color on pages 5, 6 and 7 of the revised manuscript. Please note that we have also rewritten all the theorems to make them more clear.
>
> 5) I don't think the author provides a completely rigorous justification for the collapse phenomenon.
>
> Answer: Our theorems are mathematically rigorous but perhaps the exposition was not, so we tried our best to revise it in the new manuscript. Hence, we have listed all the assumptions in each theorem explicitly and state clearly now for which conditions our results are complete.
>
> 6) Eq (2) in page 4 is not trivially correct. The expectation operator (w.r.p. to lower layers) is moved into the activation function phi, justification is needed for this step.
>
> Answer: Thanks for your suggestion. We agree with the reviewer that the equation is not trivially correct and should be justified. The proof can be found in the appendix of [4], which we have included as Appendix A in the revised manuscript.
>
> 7) Theorem 4: when the Lebesgue measure of $\Omega$ is NOT finite, it is unclear how to define a uniform probability distribution on it.
>
> Answer: We agree with the reviewer that uniform distribution cannot be defined when the measure is not finite. In fact, Theorem 4 does not require that the distribution is uniform distribution, and it only requires that the expectation or the median is finite. We have modified the theorem based on the reviewer’s comment.
>
> 8) Theorem 4: the integrability assumption on y should depend on the loss: for L2 loss (L1 loss), squared (absolutely) integrable should be used. They are not the same.
>
> Answer: We agree with the reviewer that the assumption on y depends on the type of the loss. We have modified the theorem based on the reviewer’s comment.
>
> References
>
> [1] M. Raissi, P. Perdikaris, and G. Karniadakis. Physics-Informed Neural Networks: A Deep Learning Framework for Solving Forward and Inverse Problems Involving Nonlinear Partial Differential Equations. Journal of Computational Physics, 2018.
> [2] S. Rudy, J. Kutz, and S. Brunton. Deep learning of dynamics and signal-noise decomposition with time-stepping constraints. arXiv preprint arXiv:1808.02578, 2018.
> [3] B. Hanin and M. Sellke. Approximating continuous functions by relu nets of minimal width. arXiv preprint arXiv:1710.11278, 2017.
> [4] B. Poole, et al. Exponential expressivity in deep neural networks through transient chaos. NIPS. 2016.

---

> > ### Comment · AnonReviewer3 · 2018-11-28
> > **Response to aurthors (1)**
> >
> > >>>>
> > 6) Eq (2) in page 4 is not trivially correct. The expectation operator (w.r.p. to lower layers) is moved into the activation function phi, justification is needed for this step.
> >
> > Answer: Thanks for your suggestion. We agree with the reviewer that the equation is not trivially correct and should be justified. The proof can be found in the appendix of [4], which we have included as Appendix A in the revised manuscript.
> > >>>>
> >
> > The proof in the appendix A (in [4] as well) is not mathematically rigorous.
> > 1. It requires taking the width to infinity in order to use the weak law of large numbers (also bounding the moments of random variables.)  So the argument to prove Eq. (2) does not work for finite width. The infinite width requirement is not stated near eq(2).
> > 2. The order of taking the limits of the widths to infinity needs to be considered (sequentially, or simultaneously?)  In particular, the proof can be technical if simultaneously limit is being used. (Assume you want to make the arguments mathematically correct. see https://openreview.net/forum?id=HJej3s09Km) In particular, moving the limit into the *expectation* operator is not trivial.
> >
> > For relu, eq (2) is true in the finite width setting because of the symmetry of relu ( i.e.  relu(x)+relu(-x) = x; See the appendix of [3]), but it can be false for other activation functions.

---

> > > ### Author Response · Authors · 2018-11-29
> > > **Response to Reviewer 3**
> > >
> > > We thank the reviewer for the helpful comments. In our proof, we take the width to infinity, and we will revise our main text to state this requirement. We will also cite the paper the reviewer pointed out for the more rigorous proof, and mention that Eq. (2) is true for ReLU in the finite width setting.

---

> ### Author Response · Authors · 2018-11-26
> **Response to Reviewer 3 (Part I)**
>
> We would like to thank reviewer 3 whose critical comments helped us improve our manuscript. Our paper includes both rigorous theory as well as simulations. We have tried many other cases, including solution of partial differential equations, where this collapse phenomenon may occur as we explain below. We believe that our paper is very useful as it provides, for the first time, a priori knowledge for designing safe ReLU networks. Below, we provide specific answers to the reviewer’s questions. Please note that all the edits for responses to all reviewers are marked in blue in the revised manuscript.
>
> 1) The analysis works in a very limited setting, works for ReLU but not other activations: tanh, erf, SELU etc.
>
> Answer: Thanks for your comments. We do agree that our work is focused on ReLU. There is a plethora of excellent papers in the recent literature analyzing NN with the ReLU function only. In fact, ReLU is now favored over other activation functions. Before 2010 the two commonly used non-linear activation functions were the logistic sigmoid and the hyperbolic tangent, however, the deep neural networks with these two activations are difficult to train. In 2011, ReLU was proposed, which avoids the vanishing gradient problem because of its linearity, and also results in highly sparse NNs. Since then, ReLU is favored in many deep learning models. Thus, in this study, we focus on the ReLU activation. We will extent our work to the analysis of other activation functions in our future work.
>
> 2) Very shallow networks are not popular in practice. Width>=100 is popular for fully-connected layers.
>
> Answer: Thanks for your comment. However, to our understanding, perhaps the reviewer means “very narrow networks” instead of “very shallow networks”? The selection of width depends heavily on the application. In particular, there are several emerging application areas for which deep and narrow NNs are necessary. To this end, we believe that our paper is very valuable for the following reasons. First, although the probability of NN to collapse decays fast with width, this collapse phenomenon does exist as a failure mode of NN and has never been analyzed before. Second, recently there have been a lot of emerging applications where NN is used to approximate functions and their derivatives, e.g., for solving partial differential equations [1] and dynamical systems [2]. In these applications, to approximate any continuous function with input dimension d_in and output dimension d_out, the NN width can be chosen as small as d_in + d_out [3], which could be small. For example, in the Kuramoto-Sivashinsky equation in one spatial dimensional, d_in = 2 and d_out = 1, therefore the width of NN can be 3. However, in [1] the partial differential equation is encoded in a separate coupled NN with the un-informed NN, and the depth increases depending on the order of the derivative, which for the Kuramoto-Sivashinsky equation involves a fourth-order derivative. If we start with a depth of 10 the final depth expands to 160. If we use the minimum width of 3 suggested by the theory of [3], this would lead to a total collapse of NN. In fact, our theory and Fig. 6 (right) show that to avoid collapse we have to choose a NN with width of about 15, which is 5 times higher than the value recommended by [3]. Therefore, our research is very useful for the design of narrow NNs in these applications as it gives the minimum width necessary to avoid collapse of the network. More broadly, having obtained this first theoretical result that provides a priori knowledge for the safe design of NN will spark development of other theoretical works for more general NN with diverse activation functions and other conditions. In fact, one of our colleagues at Brown University has already started such theoretical work, which is ongoing and is focusing on other non-symmetric distributions.

---

### Official Review · AnonReviewer2 · 2018-11-02
**resutls are simple, interesting but seem not very helpful in practice**

**Rating:** 6
**Confidence:** 4

**Review:**

This paper shows that the training of deep ReLU neural networks will converge to a constant classifier with high probability over random initialization (symmetric weight distributions) if the widths of all hidden layers are too small.

Overall, the paper is clearly written. I like the main message of the paper and the simplicity of its analysis. To some extent, I think that the results could add to our current understanding of the limitations of deep narrow networks, both theoretically and practically.

On the other hand, my main concern at the moment is that the results seem to be informative only for low dimensional data and networks of small width. In particular, the bound on depth in eq (5) scales too fast with width. Figure 6 shows that with width 16 the bound on depth is already too loose that it could be of any use in practice.

Other comments and questions:
In Figure 6+7, it's not clear how many times each experiment is repeated in order to get the numerical estimations of probabilities, and which exactly weight distributions are used here?

The statement of Theorem 1 and its proof looks a bit suspicious to me. This theorem first makes an assumption on a given network with fixed weights, but then makes some statement about random weights...This apparently does not make much sense to me because a given network has nothing to do with random weights, but the current proof is actually using the assumption made on the given network as a constant classifier to prove the probabilistic statement. I hope to see some clarification here.

It would be interesting to discuss the results of this paper with recent work [1,2] which also studied deep narrow networks but from other perspectives:
[1] Neural networks should be wide enough to learn connected decision regions. ICML 2018
[2] The Expressive Power of Neural Networks: A View from the Width. NIPS 2017

---

> ### Author Response · Authors · 2018-11-26
> **Response to Reviewer 2 (Part II)**
>
> 4) It would be interesting to discuss the results of this paper with recent work which also studied deep narrow networks but from other perspectives.
>
> Answer: We thank the reviewer for providing the two references. We have discussed these articles in the introduction. Please see more details highlighted in the blue color on page 2 of the revised manuscript.

---

> ### Author Response · Authors · 2018-11-26
> **Response to Reviewer 2 (Part I)**
>
> We would like to thank reviewer 2 whose critical comments helped us improve our manuscript. We believe that our paper is very useful as it provides, for the first time, a priori knowledge for designing safe ReLU networks which are used a lot in practical application recently. Below, we provide specific answers to the reviewer’s questions. Please note that all the edits for responses to all reviewers are marked in blue in the revised manuscript.
>
> 1) My main concern at the moment is that the results seem to be informative only for low dimensional data and networks of small width. In particular, the bound on depth in eq (5) scales too fast with width. Figure 6 shows that with width 16 the bound on depth is already too loose that it could be of any use in practice.
>
> Answer: We believe that this research is very valuable for the following reasons. First, this collapse phenomenon does exist and has never been analyzed before, although the collapse probability decays fast with width. The analysis of such unfavorable case will contribute to NN theories and benefit the design of NN. Second, recently many important applications start to use NN to approximate functions and their derivatives, e.g., for solving partial differential equations [1] and dynamical systems [2]. In these applications, to approximate any continuous function with input dimension d_in and output dimension d_out, the NN width can be chosen as small as d_in + d_out [3], which could be small. For example, in the Kuramoto-Sivashinsky equation in one spatial dimensional, d_in = 2 and d_out = 1, therefore the width of NN can be 3. However, in [1] the partial differential equation is encoded in a separate coupled NN with the un-informed NN, and the depth increases depending on the order of the derivative, which for the Kuramoto-Sivashinsky equation involves a fourth-order derivative. If we start with a depth of 10 the final depth expands to 160. If we use the minimum width of 3 suggested by the theory of [3], this would lead to a total collapse of NN. In fact, our theory and Fig. 6 (right) show that to avoid collapse we have to choose a NN with width of about 15, which is 5 times higher than the value recommended by [3]. Therefore, our research is very useful for the design of narrow NNs in these applications as it gives the minimum width necessary to avoid collapse of the network. More broadly, having obtained this first theoretical result that provides a priori knowledge for the safe design of NN will spark development of other theoretical works for more general NN with diverse activation functions and other conditions. In fact, one of our colleagues at Brown University has already started such theoretical work, which is ongoing and is focusing on other non-symmetric distributions.
>
> 2) In Figure 6+7, it's not clear how many times each experiment is repeated in order to get the numerical estimations of probabilities, and which exactly weight distributions are used here?
>
> Answer: We thank the reviewer for pointing out the missing details. Each numerical estimation is calculated from 1 million samples of random initialization. Proposition 10 and our numerical simulations show that the probability of NN to collapse is the same for any symmetric nonzero distribution, even if different distributions are used for different weights. We have supplemented the details in our manuscript accordingly.
>
> 3) The statement of Theorem 1 and its proof looks a bit suspicious to me. This theorem first makes an assumption on a given network with fixed weights, but then makes some statement about random weights...This apparently does not make much sense to me because a given network has nothing to do with random weights, but the current proof is actually using the assumption made on the given network as a constant classifier to prove the probabilistic statement. I hope to see some clarification here.
>
> Answer: We apologize for the confusion of the statement. The weights are randomly sampled from some distributions. Different values of weights correspond to different neural network functions, so we can think of an ensemble of NNs. In the revised version, we state in the theorem that given the assumptions A1 and A2, also assuming that the output of the neural network is a constant function, then there exists a layer such that the output of this layer is 0 with probability 1.
>
> References
>
> [1] M. Raissi, P. Perdikaris, and G. Karniadakis. Physics-Informed Neural Networks: A Deep Learning Framework for Solving Forward and Inverse Problems Involving Nonlinear Partial Differential Equations. Journal of Computational Physics, 2018.
> [2] S. Rudy, J. Kutz, and S. Brunton. Deep learning of dynamics and signal-noise decomposition with time-stepping constraints. arXiv preprint arXiv:1808.02578, 2018.
> [3] B. Hanin and M. Sellke. Approximating continuous functions by relu nets of minimal width. arXiv preprint arXiv:1710.11278, 2017.

---

### Meta-Review · Area_Chair1 · 2018-12-16
**Contributes to understanding of limitations in narrow nets, but restrictive analysis**

**Confidence:** 4
**Recommendation:** Reject

**Metareview:**

The paper studies difficulties in training deep and narrow networks. It shows that there is high probability that deep and narrow ReLU networks will converge to an erroneous state, depending on the type of training that is employed. The results add to our current understanding of the limitations of these architectures.

The main criticism is that the analysis might be very limited, being restricted to very narrow networks (of width about 10 or less) which are not very common in practice, and that the observed collapse phenomenon can be easily addressed by non symmetric initialization.

There were some issues with the proofs that were covered in the discussed between authors and reviewers. The revision is relatively extensive.

This is a borderline case. The paper receives one good rating, one negative rating, and a borderline accept rating. Although the paper contributes interesting insights to a relevant problem that clearly needs contributions in this direction, the analysis presented in the paper and its applicability in practice seems to be very restrictive at this point.